# Neural Correlates of Aberrant Salience and Source Monitoring in Schizophrenia and At-Risk Mental States—A Systematic Review of fMRI Studies

**DOI:** 10.3390/jcm10184126

**Published:** 2021-09-13

**Authors:** Joachim Kowalski, Adrianna Aleksandrowicz, Małgorzata Dąbkowska, Łukasz Gawęda

**Affiliations:** Experimental Psychopathology Lab, Institute of Psychology, Polish Academy of Sciences, ul. Jaracza 1, 00-378 Warsaw, Poland; jkowalski@psych.pan.pl (J.K.); aaleksandrowicz@psych.pan.pl (A.A.); mdabkowska@psych.pan.pl (M.D.)

**Keywords:** aberrant salience, source monitoring, psychosis, cognitive biases, self-disturbance, neural, fMRI

## Abstract

Cognitive biases are an important factor contributing to the development and symptom severity of psychosis. Despite the fact that various cognitive biases are contributing to psychosis, they are rarely investigated together. In the current systematic review, we aimed at investigating specific and shared functional neural correlates of two important cognitive biases: aberrant salience and source monitoring. We conducted a systematic search of fMRI studies of said cognitive biases. Eight studies on aberrant salience and eleven studies on source monitoring were included in the review. We critically discussed behavioural and neuroimaging findings concerning cognitive biases. Various brain regions are associated with aberrant salience and source monitoring in individuals with schizophrenia and the risk of psychosis. The ventral striatum and insula contribute to aberrant salience. The medial prefrontal cortex, superior and middle temporal gyrus contribute to source monitoring. The anterior cingulate cortex and hippocampus contribute to both cognitive biases, constituting a neural overlap. Our review indicates that aberrant salience and source monitoring may share neural mechanisms, suggesting their joint role in producing disrupted external attributions of perceptual and cognitive experiences, thus elucidating their role in positive symptoms of psychosis. Account bridging mechanisms of these two biases is discussed. Further studies are warranted.

## 1. Introduction

Cognitive biases are identified as an important factor in the development and sustainment of symptoms of schizophrenia spectrum disorders (SSD) and positive symptoms in particular [1]. In addition, numerous studies link cognitive biases with the clinical and non-clinical risk of psychosis [2,3,4,5]. The contribution of the cognitive biases to positive symptoms have been confirmed by clinical studies showing a beneficial impact reduction of cognitive biases on the severity of symptoms [6]. There is no single cognitive bias that is responsible for the development of psychotic symptoms [7]. To date, several cognitive biases have been found to be related to hallucinations [8,9,10] and delusions [11,12,13]. It is then likely that several cognitive biases play a role in the etiology of psychotic symptoms. In line with cognitive approaches to psychosis [14,15] as well as with the observations that different factors may have additive effect on the risk of psychosis [16,17], different combinations of cognitive biases may also play a role. And yet, various cognitive biases are relatively rarely investigated simultaneously, despite the fact that such studies could advance understanding of how these biases interact and how they lead to the development of psychotic symptoms in combination. Such studies could help to find answers for another vital question—whether cognitive biases share similar mechanisms. One of the proposed levels of understanding of the mechanisms of cognitive biases is brain functioning [18,19,20]. In the present review, we summarised the results of studies on functional and structural neuroimaging results concerning aberrant salience and source monitoring biases, i.e. two cognitive biases that have a prominent role in theoretical accounts [14,21] and empirical studies of psychotic symptoms. We aimed to try to elucidate their shared and unique brain correlates.

### 1.1. Role of Aberrant Salience in Psychosis

On a phenomenological level, aberrant salience is described as interpreting irrelevant as relevant, highly familiar as novel, having sharpened senses, keenness for things normally deemed negligible or a sense of newly found insight or understanding [18,22]. The persistent state of heightened and aberrant salience leads to the formation of delusional beliefs as a way to make sense of meanings found in this state. In his recognized paper, Kapur [18] has conceptualized “psychosis as a state of aberrant salience” giving priority to the process of searching the (ir)relevant meanings of stimuli. On a perceptual level, aberrant salience is associated with strong prior predictions or underweighted prediction error, which leads to excessive reliance on priors and undervaluation of sensory input [23]. This account explains how aberrant salience, understood as one of the disrupted precision mechanisms, contributes to positive symptoms of psychosis, e.g. delusions [24] and auditory hallucinations [25]. However, this model is still being developed, with important questions yet to be answered [23]. One of the ongoing discussions is the issue of the conflicting evidence pointing to the role of weak and strong priors alike in the formation of positive psychotic symptoms [26].

On the behavioural level, one of the most prominently used markers of aberrant salience is a speeding of responses to irrelevant stimuli in undirected or probabilistic motivational tasks, like the Salience Attribution Test. Paradigms using different types than reward salience, like salience associated with novelty [27], are much less popular. On the self-description level, aberrant salience is consistently linked with psychotic symptoms [28,29]. For a review of studies on non-clinical populations, see [2]. However, the results of behavioural studies are ambiguous in this regard. Studies show greater implicit and explicit aberrant salience in SSD [30,31,32] and UHR individuals [33,34] in comparison to healthy controls. Notably, these studies are inconsistent in obtained effects in different types of aberrant salience—either implicit or explicit. Some studies could not confirm differences between clinical and healthy control groups at all [35,36,37]. Inconsistency in reports of behaviorally measured aberrant salience is an argument for a systematic review of such studies. In the current study, behavioural effects will be discussed along with reviewing neuroimaging results.

### 1.2. Role of Source Monitoring in Psychosis

Source monitoring is another cognitive bias that has been early linked to psychotic symptoms in schizophrenia in theoretical accounts [38,39], which gained a lot of interest over the past decades as one of the cognitive biases that could provide a potential insight into the mechanisms in the development of psychosis. It refers to a set of processes involved in making attributions about the origins of memories, knowledge, beliefs, and perceptual experiences [39]. According to the source-monitoring framework proposed by Jonhson and colleagues [39], source monitoring is based on characteristics of perception and memories in combination with the judgment process and consists of different types of discrimination. There are at least three types of source monitoring: external source monitoring, internal source monitoring, and external-internal reality monitoring. Source monitoring processes serve to distinguish between various types of cues associated with different types of sources: sensory/perceptual information, contextual (spatial or temporal) information, semantic detail, affective information, and cognitive operations.

Bentall and colleagues [40] hypothesized that reality monitoring errors might play a role in the development of hallucinations. When patients with active hallucinations are uncertain of the source of a perceived event, they have a tendency to attribute internal events to an external source. Since then, there have been a growing number of studies investigating its role in psychotic symptoms (for reviews [5,41]) and ultra-high risk states for psychosis [3].

A recent meta-analysis [41] on behavioural results showed that patients with schizophrenia have a significantly greater tendency to misperceive inner states as originating from external sources and it has been more prominent in patients with auditory hallucinations. Importantly, the effect is evident across studies, regardless of the adopted paradigm (action modality, time delay, and design). Furthermore, the tendency in patients with schizophrenia to misidentify their own speech as that of another person appears to be related to delusions and positive symptoms generally [42].

### 1.3. Possible Integration of Aberrant Salience and Source Monitoring Deficits and Aims of this Review

There are not many theoretical and empirical accounts that integrate aberrant salience and disrupted source monitoring into a comprehensive model of positive symptoms of psychosis. Griffin and Fletcher [43] presented an account joining source monitoring and predictive processing in a hierarchical model. In the predictive coding account of psychoses [23,25,44] aberrant salience is seen as a type of disruption in precision-weighing mechanisms. Proneness to strong prior beliefs (or underweighted prediction error [23]) about the experiences and expected shape of reality results in a top-down influence that takes precedence or outweighs perceptual input, resulting in false inferences. The resulting prediction errors and biased response criteria are the basis for source monitoring processes, which are taking place in suboptimal circumstances, leading to greater source monitoring bias and deficit [43]. In the minimal self-disturbance account [22,45,46,47], aberrant salience and source monitoring deficits are conceptualized as neurocognitive processes underlying a fragile or unstable minimal self. On a phenomenological level, this translates to a disturbed sense of ownership of cognitive processes and the body and a disturbed sense of agency (as in attributing own actions to oneself). However, preliminary empirical studies have found that source monitoring deficits may have more linkages to self-disturbances at very early clinical manifestations of the risk of psychosis, as compared to aberrant salience [46]. Both presented accounts assume that aberrant salience and deficits in the source monitoring are complementary and lead, more or less directly, to positive psychotic symptoms. 

The aim of the current article was to systematically review fMRI studies investigating neuronal correlates of source monitoring and aberrant salience in schizophrenia and clinical high risk states of psychosis. The main focus was to better understand their brain-level characteristics and to investigate whether these two cognitive biases may share neuronal correlates. We conducted a systematic search of comprehensive databases to find fMRI studies of these biases in said groups. We extracted and analysed behavioural and neuroimaging results, mainly considering the effects of comparative analyses between clinical and control groups, as an indicator of specific brain correlates of the biases. Additionally, we also systematically analysed other results presented by authors, like the correlational analyses of symptom severity or clinical measurements with a BOLD response to a certain bias or bias-related within-group effects.

## 2. Materials and Methods

### 2.1. Literature Search Strategy

Our review was conducted in accordance with PRISMA 2020 guidelines for conducting and reporting systematic reviews [48]. The search was a part of a larger project on systematically reviewing associations between schizophrenia or psychosis risk and various cognitive biases. The large project was pre-registered at the Center for Open Science registry (https://osf.io/hu97e, accessed 20 August 2021). We systematically searched four comprehensive medical and psychological databases: PubMed, EMBASE, Scopus, and PsycInfo. The search was conducted up to the end of January 2021, without specific criteria for a publication date. Articles in press were also considered if they were accepted for publication and available online.

The search was conducted by combining keywords regarding schizophrenia, risk of psychosis and keywords related to aberrant salience or source monitoring. Searched strings were as follows. For schizophrenia and risk of psychosis: (psychosis OR psychotic OR schizophrenia OR schizophrenic OR UHR OR “ultra-high risk” OR “ultra-high-risk” OR “ultra high risk” OR CHR OR “clinical high risk” OR “clinical high-risk” OR ARMS OR “at-risk mental state” OR “at risk mental state” OR “at risk mental states” OR “at-risk mental states” OR “psychosis risk” OR “risk of psychosis” OR “schizophrenia risk” OR “risk of schizophrenia” OR “risk of schizophrenic” OR “schizophrenic risk”). For aberrant salience: (“aberrant salience” OR “abnormal salience” OR salience OR salient OR “salience dysregulation” OR “disrupted salience” OR “incentive salience” OR “salience attribution test” OR “salient stimuli” OR “emotional salience”). For source monitoring: (“source monitoring” OR “source-monitoring” OR “self-monitoring” OR “self monitoring” OR “externalizing bias” OR “self recognition” OR “reality monitoring” OR “source memory” OR “source-monitoring processes” OR “source monitoring processes” OR “internal source monitoring” OR “false memories”). 

A complementary reverse search was performed. We data-scraped references of found studies, removed duplicates, and manually examined these references for any papers not identified by the computerised literature search.

### 2.2. Literature Selection

We deemed studies suitable for systematic review if they met the following inclusion criteria. First, we included only studies assessing cognitive biases with experimental tasks, measuring indicators like reaction times (RTs) or error rates. Second, we included only studies investigating individuals with schizophrenia, schizophrenia spectrum disorders (e.g., schizoaffective disorder, delusional disorder), at-risk mental states, or clinical high risk of psychosis. In addition, these conditions were to be assessed by a clinician (opposed to a so-called psychometric diagnosis). We did a broad search for comparative, correlational, and longitudinal studies. Third, for the aberrant salience bias, studies should employ tasks measuring aberrant salience where meaning or saliency of the stimuli was undirected, subjected to a degree of probability, or participants were blinded as for the function of cues. This excluded simpler paradigms of reinforcement learning, like monetary incentive delay task [49,50]. Fourth, for the source monitoring bias, we included only these studies that used the task directly targeting the recognition of the source of information (source monitoring decision, e.g., self-other recognition). Studies that were focused on different patterns of neuronal activation during the processing of self vs. other with no source monitoring decision performance [51,52] were excluded from the review. Fifth, we only included studies in English.

Results of the database search were scanned by title and abstract at first to exclude articles without a connection to the review topic. The second search identified articles that were relevant for the review. The second search was conducted independently by two researchers, J.K. and M.D. for aberrant salience and A.A. and Ł.G. for source monitoring. Any inconsistencies in the selected papers were resolved by a discussion.

### 2.3. Computation, Interpretation of Results, and Bias Assessment

A large proportion of researchers do not provide estimates for interpretable effect size. Where possible, we calculated or estimated an effect size for significant results and presented them as Cohen’s d’s. For interpretation of effect size, we choose cut-off points proposed by Sawilowsky [53]. The second calculator from the Psychometrica.de site (https://www.psychometrica.de/effect_size.html; accessed date: 30 May 2021) was used whenever possible. When authors did not provide suitable data, calculators 5 and 6 were used to estimate effect sizes from values of statistical tests. To broaden our search, we also decided to include and describe results with p values that were borderline significant, i.e. 0.10 > *p* > 0.05. We treated between-groups comparisons in experimental tasks of aberrant salience and source monitoring as results of primary interest, and all other types of analyses, like correlational analyses, as secondary ones. Found studies were also assessed in terms of bias. Initially we registered the bias assessment procedure with the use of QUIPS [54]. However, this tool, which is primarily designed for clinical prognostic studies, was rather ill-fitted for assessment of bias in mostly cross-sectional studies in psychopathology. We decided on creating our own template for bias assessment with consideration of the factors we deemed important in the reviewed type of studies. Tables with bias assessment are available in Appendix A. 

## 3. Results

### 3.1. Results of Systematic Review

A total of 1363 unique records for aberrant salient articles search were identified. After removing reviews, articles without an experimental measure of aberrant salience, not in English, without participants with schizophrenia or risk of psychosis, a total of 18 articles remained. Among these articles, we identified 7 fMRI studies of aberrant salience. Reverse search resulted in an additional study, giving 8 in total. Details of the studies search and selection process is presented in Figure 1a. Of these, 7 studies were comparative and cross-sectional and one study was a longitudinal study with a mean of 17 months between measurements. Four studies concerned individuals with schizophrenia or SSD (one of these studies also concerned a group of healthy controls with high scores on a delusions inventory), 1 study concerned patients with FEP and risk of psychosis, and 3 studies concerned individuals with risk of psychosis. Five studies employed the Aberrant Salience Test. Three other used Implicit Salience Paradigm, Salience Integration Task, and an Implicit Salience Attribution Task. Seven studies employed fMRI for aberrant salience measures and one study employed fMRI for measurement of self-referential processing, which was then correlated with the aberrant salience measure.

A total of 1083 unique records for source monitoring articles search were identified. After removing reviews, articles without an experimental measure of source monitoring, not in English, without participants with schizophrenia or risk of psychosis, a total of 100 articles remained. Among these articles, we identified 8 fMRI studies of source monitoring or self-monitoring. Reverse search resulted in three additional studies, giving 11 in total. Details of the studies search and selection process is presented in Figure 1b. Of these, 10 studies were comparative and cross-sectional and one study was an intervention study, where intensive training of cognitive functions was administered with 6 months between measurements. Ten studies concerned individuals with schizophrenia or SSD and 1 study concerned patients with FEP. We decided to include all types of source monitoring paradigms in the current review due to proven consistency in results reported in the previous meta-analysis on source monitoring [41]. Five studies used the source monitoring paradigm with instant discrimination of the source of the speech. The remaining 6 studies implemented the source monitoring task with an encoding phase and source memory identification phase with between 6 to 45 min delay. In most of the paradigms with two phases, participants underwent fMRI scanning only during the second phase with retrieval. In all of those paradigms, the task was to discriminate between perceived/imagined or/and self/other generated stimuli or discriminate between different forms of stimuli presentation (self/other speech, sentences, word puzzles, pictures or pictures with labels etc.). Six studies used auditory stimuli or auditory stimuli accompanied with visual cues, in forms of either pre-recorded speech or speaking while undergoing fMRI. Additionally, one study manipulated participants expectancies using visual cues that were either congruent or incongruent with the speech as well as ambiguity (discrimination between distorted and undistorted speech).

### 3.2. Aberrant Salience

Table 1 presents details of 8 fMRI articles on aberrant salience. 

#### 3.2.1. Behavioral Results for Aberrant Salience

As for implicit aberrant salience (IAbS), only 3 of 8 studies showed significant differences between clinical and control groups [31,34,55]. The study by Katthagen and colleagues showed significantly higher IAbS in SCH patients than HC with a small effect size. The study by Pankow and colleagues [31] showed a significantly higher IAbS in SCH patients than HC with medium effect size, but not significantly higher than subclinical delusions group, although with a relatively similar medium effect size. The longitudinal study by Schmidt and colleagues [34] showed a significantly higher IAbS in UHR participants across both measurements with a large effect size. Although, this effect was only significant in the second measurement, where the difference between groups was very large in effect size and driven by a drop of IAbS score in the HC group compared to the first measurement.

For explicit aberrant salience (EAbS), only 2 of 5 studies reporting this parameter showed significant differences between clinical and control groups [33,34] and one on a trend level [56]. In a study by Roiser and colleagues [33], UHR participants had higher EAbS than HC with a large effect size. In the study by Schmidt and colleagues [34], UHR participants had higher EAbS with a medium effect size in the first measurement but not in the second. The study by Smieskova and colleagues [56] showed a trend-level result for comparing four groups—HC, ARMS, FEP, with and without medication. Interestingly, in this study, HC had the second-highest indicator of EAbS. Comparison of the two extreme groups, FEP with medication and ARMS, showed a difference with a large effect size.

#### 3.2.2. Behavioral Results for Adaptive Salience

Adaptive salience is defined as a speeding of responses for relevant stimuli relative to irrelevant stimuli [30] and is often measured along aberrant salience. For implicit and explicit adaptive salience (IAdS and EAdS, respectively), only 2 of 5 eligible studies using the Salience Attribution Test showed significant differences between clinical and control groups [31,34]. Notably, a study by Walter and colleagues [36] used Salience Attribution Test but reported only results for aberrant salience. In the Study by Pankow and colleagues [31,34], subclinical delusions group had significantly lower IAdS in comparison to HC with a large effect size but not in comparison to the SCH group (small effect size). There was also a medium, but not significant, difference between SCH group and HC. Additionally, the SCH group showed significantly decreased EAdS in comparison to HC and subclinical delusions groups, both with a very large effect size. Schmidt and colleagues [34] showed significantly lower IAdS in UHR participants with a very large effect size across both measurements. The difference in the first measurement was significant and very large in effect size, and the difference in the second measurement was at a trend level with a medium effect size. In EAdS, the results showed a significant difference with a large effect size across both measurements—the difference in the first measurement was significant and very large in effect size, and the difference in the second measurement was not significant with a medium effect size.

#### 3.2.3. Neuroimaging Results for Group Differences in Salience and Reward Prediction

For group differences in the BOLD responses, there were 6 studies that reported such analyses. The study by Katthagen and colleagues [55] focused on modelling subjective relevance and did not present group comparisons in fMRI measurement. The study by Pankow and colleagues [31] used fMRI for a self-referential processing task, the results of which were correlated with the results of the Aberrant Salience Test. Of these 6 studies, 4 showed significant group differences. Studies employing SAT used two contrasts of interest. Aberrant reward prediction—contrast of irrelevant cues assessed as high-probability of reward vs. those assessed as low-probability of reward and adaptive reward prediction—contrast of relevant cues with high-probability of reward vs. those with low-probability of reward [57], though they can be described as aberrant and adaptive salience contrasts [56]. The study by Walter and colleagues [36] showed that aberrant salience contrast was associated with greater BOLD response in L-insula in the SCH group with higher levels of positive symptoms. In the study by Smieskova and colleagues [56], there were two significant differences in BOLD response in the aberrant salience condition. FEP patients without medication showed higher activation in comparison to HC in the R-cuneus and R-middle occipital gyrus, and HC showed higher activation in comparison to ARMS and FEP in the L-inferior parietal lobule. Numerous effects for adaptive salience from this study are detailed in Table 1. Study by Schmidt and colleagues [34] found no significant effects for BOLD responses associated with aberrant reward prediction. However, they found significant effects for adaptive reward prediction—lower BOLD responses in UHR group’s ventral striatum, calcarine sulcus and midbrain bilaterally and in the L-cuneus and L-middle temporal gyrus across both measurements. In the first measurement, such an effect was found in the ventral striatum bilaterally and the L-parahippocampal and L-middle temporal gyrus and cerebellum; and in the second measurement in both ventral striata. In the study by Winton–Brown and colleagues [27], there was a higher bold response for reward prediction cues in the UHR group in L-ventral pallidum and L-midbrain.

#### 3.2.4. Additional Neuroimaging Results 

Besides group comparisons, all of the studies report other types of analyses, like correlations between BOLD response from aberrant salience and other clinical features, e.g., psychopathological symptoms. Details of these results are presented in Table 1. 

Interestingly, apart from main effects in the salience network regions reported in a paper by Smieskova and colleagues [56], there was a negative correlation between BOLD response in R-insula and hallucinations in a drug naive FEP group (*r* = −0.64). Another study [36] showed that BOLD response associated with aberrant reward prediction in the left posterior insula correlated negatively with the dose of antipsychotic medication. The activity of a salience-related brain structure was also associated with relevance weighted prediction error, i.e., increased activation in the anterior cingulate cortex in both clinical and control groups [55]. 

There were also several studies reporting effects in the basal ganglia, apart from the between-group effects reported in the papers of Schmidt and colleagues [34] and Winton-Brown and colleagues [27]. In the study of Roiser and colleagues [33], there were associations of dopamine synthesis capacity in the ventral striatum and adaptive reward prediction in caudate—positive in HC and negative in UHR, and analogous results for aberrant reward prediction signal in the hippocampus and dorsal striatal synthesis capacity. However, the authors reported only correlations for peak voxels, rendering results hard to interpret [58]. The study by Esslinger and colleagues [59] reported a difference in correlations in the bilateral ventral striatum between a monetary incentive vs. famous and non-famous faces contrasts. However, they reported only peak–voxel correlations. Additionally, FEP and HC groups did not differ in the strength of their correlations. The longitudinal study [34] reported that improvement in abnormal beliefs over time was associated with the increase in activation during adaptive reward prediction in the right ventral striatum; however, they reported correlation coefficients only for the peak voxel, undermining the reliability of the analysis [58]. Another study [27] showed that reward-induced changes in connectivity between the ventral striatum and midbrain were significantly greater in UHR than in HC. In addition, this change in connectivity correlated with CAARMS unusual thought content subscale in UHR participants.

**Table 1 jcm-10-04126-t001:** Studies on aberrant salience.

Study	Study Type	Sample Sizes	Clinical Sample(s) Characteristics	Experimental Task	Aberrant Salience Behavioural Results	Main Neuroimaging Results—Group Comparisons	Additional Neuroimaging Results
Esslinger et al., 2012 [59]	A comparative study of FEP and HC; an fMRI study	FEP = 27 HC = 27	Convenience sample from an admissions centre for a mental health hospital; never medicated	Implicit Salience Attribution Task (famous and non-famous faces)	- No significant effect reported for face x group interaction - “Three way interaction indicated that patients profited less from famousness once the stimuli were colourful (control task condition, colourful vs dull)”—Esslinger et al., 2012, p. 117.	- “There were no significant differences in activation between patients and controls regarding the contrasts of interest in the whole brain or ROI analyses”—Esslinger et al., 2012, p. 118	- A correlation in the bilateral ventral striatum ROI between activation in two tasks contrasts (monetary incentive vs control in monetary incentive task) vs (famous vs non-famous face presentation). Correlations for peak voxels revealed significant relationships between contrasts in FEP, right ventral striatum: *r* = 0.58, left ventral striatum: *r* = 0.48. Correlations in the HC were not significant. Additionally, correlation coefficients were not significantly different between groups.
Katthagen et al., 2018 [55]	A comparative study of SCH and HC; an fMRI study	SCH = 42 HC = 42	Convenience sample from inpatient and outpatient units; all participants had antipsychotic medication	Implicit Salience Paradigm	- SCH showed increased implicit aberrant salience compared to HC (*d* = 0.39) - SCH showed greater irrelevance bias (described in the paper) compared to HC (*d* = 0.55)	Not applicable	- relevance weighted prediction error (described in the paper) correlated with increased response in the anterior cingulate cortex in all participants - relevance weighted absolute prediction errors (described in the paper) correlated with decreased response in the L-hippocampus in all participants - irrelevance bias correlated with decreased relevance weighted prediction errors in bilateral nucleus accumbens response
Pankow et al., 2016 [31]	A comparative study of SCH, subclinical delusions and HC; an fMRI study	SCH = 29 subclinical delusions = 24 HC = 50	SCH: a convenience sample from a hospital department of psychiatry and psychotherapy. Most of the patients were medicated; subclinical delusions: people with results in 4Q of PDI from a large internet sample	Salience Attribution Test	- Implicit Aberrant Salience: SCH showed significantly greater IAbS in comparison to HC (*d* = 0.68), but not subclinical delusions group (*d* = 0.56 - Explicit Aberrant Salience: no significant differences - Implicit Adaptive Salience: subclinical delusions group showed decreased IAdS salience in comparison to HC (*d* = −0.85), but not SCH (*d* = −0.14); difference between SCH and HC was also not significant (*d* = −0.52) - Explicit Adaptive Salience: SCH showed significantly decreased EAdS in comparison to HC (*d* = −1.40) and subclinical delusions group (*d* = −1.24)	Not applicable	- A significant relationship between aberrant salience score in SCH related to self-referential task activation in the ventromedial prefrontal cortex (*r* = −0.60) - No such correlation was observed for subclinical delusions group and HC
Walter et al., 2016 [36]	A comparative study of SSD with lower and higher levels of positive symptoms; an fMRI study	SSD with higher positive symptoms = 21 SSD with lower positive symptoms = 21	A convenience sample of patients with FEP and chronic SCH patients with a history of violent offense; most of the patients were medicated	Salience Attribution Test	- Implicit Aberrant Salience: no significant differences - Explicit Aberrant Salience: no significant differences - Implicit Adaptive Salience: no data - Explicit Adaptive Salience: no data	- Aberrant salience was associated with increased BOLD response in left insula in the higher positive symptoms group	- Aberrant reward prediction signals in the left posterior insula correlated negatively with the chlorpromazine-equivalent doses (*r* = −0.31)
Smieskova et al., 2015 [56]	A comparative study of ARMS, FEP and HC; an fMRI study	FEP with medication = 12 FEP without medication = 17 ARMS = 34 HC = 19	A convenience sample of patients with FEP (medicated and unmedicated) and ARMS patients	Salience Attribution Test	- Implicit Aberrant Salience: no significant differences - Explicit Aberrant Salience: a trend result (*p* = 0.096) for group effect (FEP with medication > HC > FEP without medication > ARMS) (*d* for FEP with medication > ARMS = 0.88) - Implicit Adaptive Salience: no significant differences - Explicit Adaptive Salience: no significant differences	- In aberrant salience, FEP without medication showed higher BOLD response in comparison to HC in the R-cuneus and R-middle occipital gyrus - In aberrant salience, HC showed higher BOLD response in comparison to ARMS and FEP in L-inferior parietal lobule - In adaptive salience, ARMS showed lower BOLD response in comparison to HC in R-supramarginal gyrus and R-inferior parietal lobule - In adaptive salience, FEP without medication showed lower BOLD response in comparison to HC in dorsal part of L-anterior cingulate gyrus, L-middle frontal gyrus and L-precentral gyrus - In adaptive salience, FEP with medication showed lower BOLD response in comparison to HC in R-insula and R-precentral gyrus, R-paracingulate gyrus, and R-anterior cingulate gyrus - In adaptive salience, combined FEP groups showed lower BOLD response in comparison to HC in R-precentral gyrus and R-insula - In adaptive salience, FEP with medication showed lower BOLD response in comparison to ARMS in bilateral paracingulate gyri and L-anterior cingulate gyrus	- Negative correlation between R-insular adaptive salience activation and hallucinations in FEP without medication (*r* = −0.64)
Roiser et al., 2013 [33]	Comparative study of UHR and HC; an fMRI and PET study	UHR = 18 HC = 18	A convenience sample; 2 participants with antipsychotic medication (authors checked that data from these participants did not alter the results)	Salience Attribution Test	- Implicit Aberrant Salience: no significant differences - Explicit Aberrant Salience: UHR showed greater EAbS in comparison to HC (*d* = 0.88) - Implicit Adaptive Salience: no significant differences - Explicit Adaptive Salience: no significant differences	- No significant group differences	- In both groups collapsed, there was a positive relationship between aberrant reward prediction response in the ventral striatum and explicit aberrant salience, “however, the slope of the regression line was significantly flatter and nonsignificant in UHR” (Roiser et al., 2013, p. 1331) - Aberrant reward prediction in the hippocampus was positively correlated with dorsal striatal dopamine synthesis capacity in HC (*r* = 0.65) and negatively in the UHR (*r* = −0.52) in a peak voxel - Adaptive reward prediction response in the caudate was positively correlated with ventral striatum dopamine levels in HC (*r* = 0.63) and negatively in UHR (*r* = −0.63) in a peak voxel
Schmidt et al., 2017 [34]	A longitudinal study of UHR and HC; an fMRI study	UHR = 23 HC = 13	A convenience sample from a clinical service for people at high risk of psychosis	Salience Attribution Test	- Implicit Aberrant Salience: in both timepoints UHR showed significantly higher IAbS than HC (*d* = 0.93); at follow-up, the UHR group showed significantly higher IAbS (*d* = 1.25) (drop of IAbS in HC), but not at baseline - Explicit Aberrant Salience: at baseline, UHR showed significantly higher EAbS (*d* = 0.78), but not at follow-up - Implicit Adaptive Salience: in both timepoints, UHR showed lower IAdS (*d* = −1.21); at baseline significantly lower IAdS (*d* = −1.33) and at follow-up at trend level (*p* = 0.062, *d* = −0.69) - Explicit Adaptive Salience: UHR showed lower EAdS (*d* = −0.84) in both timepoints; significantly lower EAdS (*d* = −1.07) at baseline, but not at follow-up (*d* = −0.42)	- No significant effects of group or time, and no group x time interactions, significant group differences in aberrant reward prediction at either baseline or follow-up - No significant group × time interactions for adaptive reward prediction - In both timepoints, UHR showed weaker BOLD response than HC in the ventral striatum, calcarine sulcus and midbrain bilaterally and in the left cuneus and middle temporal gyrus in adaptive reward prediction - At baseline UHR showed significantly less activation in the ventral striatum bilaterally and the left parahippocampal and middle temporal gyrus, and cerebellum during adaptive reward prediction - At follow-up the UHR showed significantly less activation in the ventral striatum bilaterally during adaptive reward prediction	- No significant relationships between changes in clinical features and longitudinal changes in brain activation during aberrant reward prediction - Improvement in abnormal beliefs over time was associated with the increase in activation during adaptive reward prediction in the right ventral striatum and in the supplementary motor cortex bilaterally (peak-voxel correlation)
Winton-Brown et al., 2017 [27]	A comparative study of UHR and HC; an fMRI study	UHR = 29 HC = 32	A convenience sample from a clinical service for people at high risk of psychosis	Salience Integration Task	- No significant differences for reward, novelty and aversion conditions	- UHR showed significantly greater BOLD response to reward-predicting cues than HC in the L-ventral pallidum and L-midbrain; no areas where HC showed greater activation than UHR- No significant differences for novelty condition - No significant differences for aversion condition	- Reward-induced modulation of ventral striatum/pallidum to midbrain connectivity was significantly greater in UHR than in HC - Described above modulation was correlated with CAARMS unusual thought content in UHR (*r* = 0.50), but not other positive symptoms subscales - No significant correlations between group differences in activation related to reward processing and CAARMS positive symptoms

FEP—first episode of psychosis group, HC—healthy controls group, SCH—schizophrenia or schizophrenia spectrum disorder group, ARMS—at risk mental states group, UHR—clinical ultra-high risk of psychosis group, IAbS—implicit aberrant salience condition, EAbS—explicit aberrant salience condition, IAdS—implicit adaptive salience condition, EAdS—explicit adaptive salience condition, CAARMS—Comprehensive Assessment of at Risk Mental States.

### 3.3. Source Monitoring

Table 2 presents details of 11 fMRI studies on source monitoring. 

#### 3.3.1. Behavioral Results for Source Monitoring

Out of 11 studies considered in the current review, 10 implemented an external–internal source monitoring paradigm. Of those, most studies showed significant differences between clinical and control groups. Results [60,61] where SCH patients were divided into those with active auditory verbal hallucinations (AVH) and those that did not experience auditory verbal hallucinations in their lifetime (non-AVH) found that patients with AVH made significantly more external misattributions while performing a source monitoring task than the non-AVH patients and HC group. The study by Garrison et al. [62] showed that SCH had significantly less general correct responses than HC in reality monitoring with a large effect size. Study by Kumari and collaborators [63] also showed that the SCH group had impaired performance in source monitoring task in comparison with the HC group with a large effect size. In the study by Thoresen and colleagues [64], patients made more misattributions where they classified the imagined condition as a viewed condition with a very large effect size. Vinogradov and collaborators [65] demonstrated that the SCH group was significantly less accurate in self-generated items when compared with the HC (independently of age or poorer item recognition memory) with a very large effect size (although the sample size was relatively small). A study by Subramaniam and colleagues [66] investigated changes in the reality monitoring task performance after 16 weeks of cognitive training (participants trained cognitive functions such as basic emotion recognition, theory of mind, memory, executive functions). The results revealed that patients who took part in the training compared to patients who played computer games showed a significant improvement in their accuracy performing a reality monitoring task. Another study [67] showed that HC performed better than the SCH group at overall source-memory identification across both self-generated and externally presented word items. In another study [68], there was a significant group effect in the overall task performance accuracy, that is, patients revealed deficits in source monitoring as compared to HC. A study conducted on the FEP group [69] showed that patients made significantly more total errors than HC when listening to their own voice preceded by an invalid cue with a large effect size. A remaining study implemented an external-external source monitoring paradigm [70]. Results showed no differences in the response bias index (tendency to remember word–items as pictures) between SCH and HC groups. However, authors demonstrated that greater severity of visual hallucinations was associated with increased rates of false memories of pictures, but there was no relation with the verbal hallucination score. Results are presented in Table 2 in detail.

**Table 2 jcm-10-04126-t002:** Studies on source monitoring.

Study	Study Type	Sample Size	Sample Characteristics	Experimental Task	Behavioural Results	Main Neuroimaging Results—Group Comparison	Additional Neuroimaging Results
Allen et al., 2007 [42]	Comparative study of SCH sample and HC, an fMRI study	SCH = 20 (division on AVH = 10; non-AVH = 10) HC = 11	A convenience sample of psychiatric patients recruited through the South London and Maudsley National Health Service Trust	Source monitoring task (discrimination between self/researcher and distorted/undistorted speech )	- A significant interaction between the effects of source of speech and group - A significant group difference in the self speech condition, AVH group made significantly more misattribution errors than the participants in both the non-AVH and HC	- A significant interaction between response accuracy (correct/misattribution) and group in the left middle temporal gyrus; in both HC and non-AVH groups there was greater activation for correct responses (correct attribution of either self or other speech) than for misattributions, whereas there was no significant difference in the AVH group; similar patterns of activation occurred when only self speech condition (i.e. the correct identification of self speech v. its misattribution to an external source) was analysed	-
Garrison et al., 2017 [62]	Comparative study of SCH sample and HC, an fMRI study	SCH = 20 HC = 20	A convenience sample	Reality monitoring task (discrimination between perceived/imagined and self/researcher word-pairs)	- The SCH group had significantly less correct responses than HC (*ηp2* = 0.175; *d* = 0.92 ) - No significant interaction between group and reality monitoring condition - The SCH group had significantly slower RTs than HC (*ηp2* = 0.239; *d* = 1.12)	- No significant group differences in BOLD response - A significant three-way interaction between group, task, and region (left and right medial anterior PFC, left and right dorsolateral PFC) - A between groups comparison indicated a trend towards a significant reduction in reality monitoring activity in SCH compared with HC in the left medial anterior PFC region of interest only (*p* = 0.057)	- HC exhibited significant activity associated with the reality monitoring contrast in the left medial anterior PFC region of interest and in the left and right dorsolateral PFC regions of interest, and at a whole-brain corrected voxel-wise height threshold of *p* < 0.05, in the occipital lobe - In the SCH group, significant reality monitoring-related activity was detected only in left dorsolateral PFC - No significant correlation between the percentage signal change for each participant in any of the a priori left and right medial anterior and dorsolateral PFC voxels, for the reality monitoring contrast and the working memory contrast, for either SCH or HC
Kambeitz-Ilankovic et al., 2013 [69]	Comparative study of FEP and HC; an fMRI study	FEP = 20 HC = 20	A convenience sample of psychiatric patients recruited through Maudsley National Health Service Trust	Source attribution task (discrimination between self/other speech, ambiguity: undistorted/distorted speech and validity: valid/invalid cues)	- All interactions were nonsignificant, although there was a trend for an interaction between validity, source, and group (*p* = 0.063) - Post hoc tests revealed that FEP made significantly more total errors than HC when listening to their own voice preceded by an invalid cue (non-self-cue) (*d* = 0.8)	- A significant interaction between group, validity, and source in the right MTG and in the left precuneus; in both these regions, HC showed greater activation for self-speech relative to other speech condition during invalidly cued (but not validly cued) trials; unaltered activation during both source and validity manipulations in these regions in FEP patients - A post hoc analysis showed that in HC, relative to valid self-trials, invalid self-trial condition were associated with greater activation in the right Pc, MTG, and left insula. In FEP, there were no areas more active during invalid relative to valid trials. - Non-significant interactions between group and validity and between group, validity, source, and distortion	- A negative relationship in the FEP group between the activation in the right MTG and a severity of positive psychotic symptoms measured by PANSS positive symptoms subscale (*r* = −0.62) and the PSYRATS delusion items (*r* = −0.45)
Kumari et al., 2008 [63]	Comparative study of SCH and HC; an fMRI study	SCH = 63 HC = 20	A convenience sample	Self-monitoring task (discrimination between self/other speech and undistorted/distorted speech)	- HC showed more accurate performance than the SCH group (*d* = 1.15) - A significant group× source × distortion interaction (*d* = 0.54), but after complementing the results for percentage of correct answers the interaction changed to a trend level (*p* = 0.06) - The SCH group made more misattributions than HC (*d* = 0.98)	- The SCH group showed more activity than HC in the ventral striatum, hypothalamus, and part of the thalamus in the other > self-contrast	- Within the SCH group, greater ventral striatal-hypothalamic activity during other > self contrast correlated with higher negative symptoms PANSS score (*r* = 0.29)
Mechelli et al., 2007 [61]	Comparative study of SCH and HC; an fMRI study	SCH = 20 (division on AVH = 11; non-AVH = 10) HC = 10	A convenience sample of psychiatric patients recruited via the South London and Maudsley National Healthy Trust (SLAM)	Source monitoring task (discrimination between self/researcher and disorted/undistorted speech )	- A significant interaction between source of speech and group; post-hoc tests revealed a significant difference between groups in the self-speech condition, patients with AVH misidentified their own speech as other speech more often than both HC and non-AVH patients	- A significant effect of source on the left superior temporal cortex to the anterior cingulate that differed across the three experimental groups; intrinsic connection was stronger for alien- than self-generated speech in HC and in non-AVH patients but not in patients with AVH, who expressed the reverse trend; the effect of source on this connection was significantly stronger in HC relative to AVH and in non-AVH relative to AVH, but did not differ between HC and non-AVH; thus, it is specifically impaired in patients with AVH - A significant connection from right superior temporal cortex to the anterior cingulate that was stronger for alien- than self-generated speech in all three experimental groups, but no significant differences between the groups	-
Stephan-Otto et al., 2017 [70]	Comparative study of SCH sample and HC, an fMRI study	SCH = 23 HC = 26	A convenience sample of psychiatric patients recruited from the Parc Sanitari Sant Joan de Deu network of mental health services in Barcelona	Reality monitoring task (discrimination between pictures and picture labels)	- The response bias index Br * was equivalent in the SCH and HC groups	- Not applicable	- No significant differential activation was observed for any of the contrasts studied when the patients with visual hallucinations were compared to the other patients. The patients with verbal hallucinations presented increased bilateral activation in the thalamus and in the precuneus relative to the other patients when correctly remembering pictures (pictures remembered as pictures > pictures remembered as words) - Comparison of patients with low Br versus high Br showed a differential activation at a trend level of the left superior temporal gyrus when erroneously remembering pictures (words remembered as pictures > words remembered as words)
Subramaniam et al., 2012 [66]	An intervention study of SCH; an fMRI study	SCH (division into AT: active training and CG: computer games training groups) = 31 (15 + 14 and 2 dropouts at baseline; 13 and 12 returned 6 month later) HC = 15	A convenience sample of psychiatric patients recruited from community mental health centres and outpatient clinics	Reality monitoring task (discrimination between self-generated or externally presented words)	Baseline assessment: - a significant group x condition interaction (*d* = 0.59); driven by the HC subjects who identified significantly more self-generated items than SCH subjects (*d* = 0.84), but not more externally presented items - signal detection theoretic analysis confirmed that HC subjects performed significantly better than SCH subjects during overall source memory identification of word items (*d* = 0.66) - The effect size of the overall source memory accuracy difference between HC and SCH subjects at baseline was 0.65. Post training: - A significant group x session (baseline and after 16 weeks) interaction in d-prime scores for overall source memory identification of word items (d = 0.92); a significant group x session effect for self-generated word items (d = 0.87) but not for externally presented word items. The SCH-AT subjects, when compared to the SCH-CG subjects, identified the source of significantly more word items overall at 16 weeks compared to baseline (*d* = 1.1) and also specifically identified more self-generated items (*d* = 1.01), with a trend effect for externally presented items (*p* = 0.07). The SCH-AT subjects, when compared to the HC subjects, identified the source of more word items overall at 16 weeks compared to baseline (*d* = 0.88), identifying more self-generated (*d* = 1.01) but not more externally presented items; no differences between sessions for HC or SZ-CG subjects on overall source-memory accuracy, on self-generated items or externally presented items	Baseline assessment: - A significant group effect in mPFC activity for self-generated minus externally presented items. This group effect at baseline was driven by the HC subjects, who revealed significantly more mPFC activity for self-generated items than externally presented items when compared to the SCH-CG and SCH-AT, there was no significant difference in mPFC activity between SCH-CG and SCH-AT subjects at baseline Post training: - A significant group x session interaction in mPFC reality monitoring activity, driven by the SCH-AT subjects, who had significantly more mPFC signal activation after the intervention than the SCH-CG subjects than the HC subjects; no differences between sessions for HC or SCH-CG subjects in mPFC signal for the self-generated item minus externally presented item comparison.	- A significant correlation between mPFC signal and verbal memory scores at post-training in SCH-AT (*r* = 0.51); no significant correlations at baseline nor in the SCH-CG group after intervention - no significant correlation between mPFC activity and executive functions at post-training - A significant correlation between the level of reality monitoring signal within the a priori spherical mPFC ROI immediately after training and ratings of social functioning at the 6 month follow-up (*r* = 0.55)
Subramaniam et al., 2017 [67]	A comparative study of SCH; an fMRI study	SCH = 20 HC = 20	A convenience sample of psychiatric patients recruited from a double-blind randomized clinical trial of cognitive training in schizophrenia (ClinicalTrials.gov NCT02105779).	Reality monitoring task (discrimination between self-generated or externally presented words)	- A significant main effect of group (*d* = 2.02); no significant interaction between mood and group, task accuracy and group, or between mood and task accuracy and group - between-group contrasts revealed HC performed significantly better than SCH at overall source-memory identification across both self-generated and externally presented word items (*d* = 0.87)	- ROI contrasts showed a significant between-group differences in which HC showed greater signal activation during source monitoring task than SCH group within the mPFC ROI for both positive and neutral mood states conditions and within the left putamen ROI for positive mood states; no significant between-group signal differences during source monitoring performance in the PCC for either positive mood or neutral mood states conditions or within the left putamen ROI for neutral mood - Whole-brain analysis showed no between-group differences during negative mood states, when compared to the neutral mood condition (after FWE cluster corrections)	- When HC participants were in the negative mood, signal within the left dorsal region of the mPFC negatively correlated with externally-presented identification (*r* = −0.48), and with overall reality-monitoring performance (*r* = −0.44) - when SCH were in the negative versus neutral mood, signals in both left and right dorsal mPFC ROIs correlated with better externally-presented identification (left dorsal mPFC: *r* = 0.54; right dorsal mPFC: *r* = 0.56), and with overall reality-monitoring performance (left dorsal mPFC: *r* = 0.54; right dorsal mPFC: *r* = 0.56), despite the fact that patients did not show increased signal within these ROIs during the negative MI when compared to the neutral MI.
Thoresen et al., 2014 [64]	Comparative study of SCH sample and HC, an fMRI study	SCH = 19 HC = 20	A convenience sample	Reality monitoring task (discrimination between presented or an imagined object/scene)	- No significant difference between the groups for item recognition - A significant difference between the two groups in the imagined condition (IC); patients showed lower accuracy compared to HC (*d* = 0.62) - No significant difference in the viewed condition (VC) - Patients made more misattributions where they classified the IC as VC (*d* = 1.30 ) - The SCH group also classified IC as new to a greater extent compared to the HC (*d* = 0.79) - Main effect of group in RTs; patients were responding slower than HC (*d* = 1.45) - no significant interactions between source and group	- HC showed greater activity in left DLPFC for the contrast IC>Baseline and VC>Baseline and left hippocampus for the contrast IC>Baseline compared to the SCH group—BOLD response in the regions that displayed significant voxel-wise activation in conditions and between groups showed that both groups activated left DLPFC in IC compared to the baseline task and left hippocampus in IC compared to the baseline task	- A significant negative correlations with degree of delusions and left hippocampal activity in the IC in SCH group (*ρ* = 0.54); the relationship was still significant after controlling for dose of antipsychotic medication (*r* = 0.49), and PANSS general psychopathology symptom score, (*r* = 0.48), but not significant after controlling for Information, Digit Span, or Digit Symbol-Coding (WAIS-III subscales). - Left hippocampal activity in the VC correlated significantly with delusions, (*ρ* = 0.53) - No significant association between activity in the regions of interest and the degree of hallucinations
Vinogradov et al., 2008 [65]	Comparative study of SCH sample and HC, an fMRI study	SCH = 8 HC = 8	A convenience sample	Source memory task (discrimination between the self-generated nouns and presented nouns in the sentence completion paradigm)	- No significant interaction effects for hit rates; however there was a significant effect of condition on source memory hit rate (with age as a covariant) (*d* = 1.56); SCH group had a significantly lower hit rate for self-generated items compared with the HC (*d* = 3.04); no significant differences between SCH and HC groups on hit rates for external items; old items; or new items - no interaction effects for RTs; there was a group difference at the trend level, SCH showed longer RTs on correct identification of self-generated items (*p* = 0.08) and old items (*p* = 0.06) compared with HC group	- Between-group analyses for the contrast of self-generated > externally presented items revealed that HC had significantly greater activity than SCH subjects in right rostral mPFC and in right superior frontal gyrus; SCH had significantly greater activity than HC in scattered areas that included left supplementary motor area, occipital cortex, anterior cingulate, and basal ganglia - The SVC of mPFC on the group level revealed a significant cluster of 216 voxels for the HC, with a corrected *p* < 0.001, and no significant clusters for the SCH. For the between-group contrast of HC > SCH, the SVC revealed a significant cluster of 72 voxels with a corrected *p* < 0.001 and no significant clusters for the SCH > HC	- No significant correlation between age and percent signal change in mPFC during the self-generated condition
Wang et al., 2011 [68]	Comparative study of SCH sample and HC, an fMRI study	SCH = 23 HC = 33	A convenience sample of psychiatric patients recruited from from psychiatric hospitals and community health agencies in and around Vancouver, BritishColumbia, Canada	Self–other source monitoring task (discrimination between self-generated and other-generated words when solving a puzzle)	- No significant interaction between group and condition (*p* = 0.1) - Significant group effect was observed in performance accuracy (*d* = 0.81) - No significant interaction between group and condition and no significant group effect in RTs (correct trials only)	- Interaction analysis on the source-memory contrast of self-generated (SG) greater than other-generated (OG) found a significant between-group difference in the left superior temporal gyrus; HC subjects showed significantly higher mPFC-LSTG connectivity during OG than SG conditions, and SCH group showed significantly higher connectivity during SG than OG conditions.	-

FEP—first episode of psychosis group, HC—healthy controls group, SCH—schizophrenia or schizophrenia spectrum disorder group, AVH—auditory verbal hallucinations group, RTs—reaction times, PFC—prefrontal cortex, mPFC—medial prefrontal cortex, MTG—middle temporal gyrus, DLPFC—dorsolateral prefrontal cortex, FWE—family-wise error correction, MI—mood induction, PCC—posterior cingulate cortex, SVC—small-volume correction, * index Br—the propensity to report words as pictures in case of uncertainty.

#### 3.3.2. Neuroimaging Results for Group Differences in Source Monitoring

For the external–internal source monitoring paradigm, nine studies reported group differences in the BOLD response. The one study that implemented the external–external source monitoring paradigm did not provide group comparison in the BOLD response. The study by Allen and colleagues [60] compared brain activation during source monitoring task in patients with auditory verbal hallucinations (AVH), patients without auditory verbal hallucinations (non-AVH), and the HC group. The results showed that in HC and non-AVH patients, greater activation was seen in the middle temporal gyrus for correct responses than for misattributions, while there was no significant difference in the AVH group. Another analysis of the above-mentioned data [61] revealed that intrinsic connection between the left superior temporal and anterior cingulate cortex was modulated by source of speech in patients without AVH and healthy controls but not in patients with AVH, where the reverse trend was found. The study by Garrison and colleagues [62] showed that performance in the reality monitoring task is associated with the differential BOLD response in the medial anterior prefrontal cortex between the investigated groups. In a study by Vinogradov et al. [65], the results revealed that the SCH group manifested a deficit in the rostral medial prefrontal cortex, while undergoing a source monitoring task. On the other hand, SCH had significantly greater activation in the left supplementary motor area, occipital cortex, anterior cingulate, and basal ganglia. An intervention study by Subramaniam and colleagues [66] showed that during baseline assessment, the HC group revealed significantly more mPFC activity for self-generated items than externally presented items when compared to the patients with schizophrenia that underwent computer games condition (SCH-CG) and patients that underwent active training condition (SCH-AT). While, after intervention, SCH–AT had significantly more mPFC activity than SCH-CG, no differences were found for HC and SCH-CG. In another study [67], in addition to the previously used paradigm [65,66], mood state conditions were implemented, where positive, negative, or neutral mood states were induced prior to the task completion. Results showed differential activation of BOLD response in the right dorsal mPFC during the task performance when HC and SCH groups were compared. A study conducted by Thoresen and colleagues [64] displayed that in SCH patients reduced accuracy in the imagined condition was accompanied with reduced activity in the left dorsolateral prefrontal cortex as well as in the left hippocampus. Other results [63] demonstrated that the SCH group showed more activity than HC in the ventral striatum, hypothalamus, and part of the thalamus, when the other versus self-contrast was applied.

#### 3.3.3. Additional Neuroimaging Results

Additionally to group comparisons, 8 studies report other types of analyses, like correlations between BOLD response from source monitoring and clinical features, e.g. psychopathological symptoms or cognitive functions. Details of these results are presented in Table 2. 

In the context of clinical features, like psychopathological symptoms, one study [69] showed that severity of positive psychotic symptoms measured by PANSS positive symptoms subscale (*r* = −0.620) and the PSYRATS delusion items (*r* = −0.451) negatively correlated with the activation in the right MTG in the FEP group. Kumari and collaborators [63] conducted correlations between BOLD response during source monitoring task performance and PANSS scores. The results showed that within the SCH group, greater ventral striatal–hypothalamic activity during other versus self-contrast correlated with higher negative symptoms PANSS score (*r* = 0.29). Another study [64] demonstrated that in the patient group, there is a negative relationship between the degree of delusions and left hippocampal activity in the imagined condition (IC) (*ρ* = 0.54). Moreover, left hippocampal activity in the viewed condition (VC) correlated significantly with delusions (ρ = 0.53). Stephan–Otto and collaborators [70] showed that the patients with verbal hallucinations presented increased bilateral activation in the thalamus and in the precuneus relative to the other patients when correctly remembering pictures. On the other hand, when patients with visual hallucinations were compared to the other patients, no differences in brain activation was seen. In a study by Subramaniam and colleagues [67], in which different mood states were induced, the results showed that when HC were in a negative mood, there was a negative relationship between the left dorsal region of the mPFC and externally-presented identification (*r* = −0.48) as well as overall reality-monitoring performance (*r* = −0.44). On the other hand, when SCH were in a negative mood versus neutral mood, there was a positive relationship between signals in both left and right dorsal mPFC ROIs and better externally-presented identification (left dorsal region of the mPFC: *r* = 0.54; right dorsal region of the mPFC: *r* = 056), and overall reality-monitoring performance (left dorsal region of the mPFC: r = 0.54; right dorsal region of the mPFC: *r* = 0.56).

Subramaniam et al. [66] compared the mPFC activity with different cognitive exercises that they applied during the intervention. The results show a significant correlation between mPFC signal during the reality monitoring task and verbal memory scores at 16 weeks training in SCH-AT (*r* = 0.51). Moreover, there was a significant correlation between the level of reality monitoring mPFC signal measured immediately after training and ratings of social functioning at the six-month follow-up. Whereas, no significant relationship was found in mPFC activity and executive functions.

## 4. Discussion

In the current systematic review, we found eight fMRI studies of aberrant salience and eleven fMRI studies of source monitoring in schizophrenia and clinical risk of psychosis. As stated in the introduction, cognitive biases are rarely studied simultaneously, and the results of this review seem to confirm that. To our knowledge, there are no neuroimaging studies linking paradigms measuring aberrant salience and source monitoring. We aimed to review the neuronal correlates of aberrant salience and source monitoring in the spectrum of psychoses and elucidate specific and possible shared functional neuronal mechanisms of these cognitive biases. We also reviewed behavioural results obtained in the found studies.

Furthermore, we assessed the quality and possibility of bias of found studies. We focused on exclusion and inclusion criteria reporting, replicable and reliable assessment of cognitive biases and possible confounding factors in the analysis of the participants. The overall quality of the papers was suitable. Most of the studies measured aberrant salience and source monitoring with computerized tasks, which are highly reliable and provided suitable descriptions of their tasks, which allow for further replicability. Some of the studies did not report exclusion and inclusion criteria in the clinical groups, like age or neurological disorders. In addition, some did not take into account possible confounders in the statistical analyses, like differences in IQ of clinical and control groups.

### 4.1. Behavioral Results for Aberrant Salience

The results show that the most common paradigm was the Salience Attribution Test, with five out of eight studies using it, followed by single studies using other types of tasks. Most of the studies researched schizophrenia spectrum or/and clinical risk of psychosis compared to a non-clinical control group. One study [36] studied schizophrenia participants with high and low positive symptoms. Regardless, the obtained results were diversified, with only about one-third of the effects concerning aberrant salience, implicit or explicit, being reported as significant. These results may posit a question about reliable differences in aberrant salience between individuals on the psychosis continuum and healthy controls, as measured by behavioural tasks. Alternatively, perhaps, methods and conditions in which tests of aberrant salience were conducted could have influenced results obtained in the reviewed studies.

The three significant effects for implicit aberrant salience were varied, spanning from small (*d* = 0.39) (largest sample size, schizophrenia patients) [56], through medium (*d* = 0.68) [31] to large (*d* = 0.93) (longitudinal study in an UHR sample) [34]. The two effects for explicit aberrant salience were more consistent, with effects in the large range (*d* = 0.78–0.88) [33,34]. We estimated sample sizes required to obtain a significant and suitably powered effect in GPower [71]. We assumed effect sizes of *d* = 0.6 for implicit aberrant salience and *d* = 0.8 for explicit aberrant salience, *α* = 0.05 (two-tailed), power of *β* = 0.95 (as avoiding false negatives at this point seems crucial, [68], and a design comparing two independent groups. These analyses indicated single group sample sizes of n = 74 and n = 42 (n = 45 and n = 26 for the often assumed value of *β* = 0.80) for implicit and explicit aberrant salience, respectively. This would indicate that the majority of the fMRI studies of aberrant salience up to date were underpowered in the context of behavioural results and thus might have had problems with detecting effects of interest. Hence, future studies should consider employing larger samples.

Additionally, the effects of aberrant salience in the schizophrenia spectrum may be hard to observe, and studies on aberrant salience in schizophrenia patients may be hard to interpret. This happens in the context of dopamine antagonist medication used in the treatment of this disorder. Some studies point to the possibility that antipsychotic medication modulates reward and salience processing in the brain and that these effects may be heterogeneous, depending on the type of medication [56,72]. In the current review, in most studies, schizophrenia participants were taking antipsychotic medications. Controlling for the effects of medication in future studies seems a viable methodological consideration. Another possibility is that fMRI setting somehow influenced tasks measuring aberrant salience, rendering them not as sensitive or specific as in a typical “desk” setting. A review of all behavioural studies of aberrant salience in schizophrenia and clinical risk of psychosis is warranted to discuss the reliability of aberrant salience tasks.

### 4.2. Neuroimaging Results for Aberrant Salience

The reviewed studies show that neural processes related to aberrant and adaptive salience are linked primarily to two major brain systems, the subcortical dopamine system and the salience network. Though, it is important to notice that the insula, anterior cingulate cortex, and subcortical structures, amygdala, ventral striatum and ventral tegmental area are regarded as part of the salience network [73]. Therefore, there is a substantial anatomical overlap between the two discussed systems. Noteworthy, recent studies show that salience network functioning is associated with glutamate [74,75,76]. Nevertheless, as parts of this networks are often considered in relation to their specific functional mechanisms, we will discuss obtained results with a division into cortico-striatal-thalamic circuitry associated with dopamine signalling and salience network, as composed of the insulae and anterior cingulate cortices. 

The first of the discussed systems is a cortico-striatal-thalamic circuitry responsible for reward and salience processing, adaptive and aberrant alike, especially striatum, hippocampus, amygdala, and prefrontal cortex [57,77]. See Howes et al. [23] for a review of reward and salience processing studies in UHR participants. A meta-analysis of studies on reward processing in schizophrenia shows a general effect of hypoactivation in patients’ ventral striatum [78]. This system is associated with motivational salience and its disruption [18]. The aberrant salience model assumes that the process of developing psychosis has a basis in the dysregulation of subcortical dopaminergic pathways in the brain [18,79,80].

In the current review, only two studies [27,34] showed direct effects of group comparisons in the mesolimbic pathway, both concerning UHR participants. One study [34] revealed diminished neural response in subcortical brain regions, ventral striatum, and parahippocampal gyrus, and the other [27] showed an increased response in the ventral striatum. However, correlational results also found effects associated with these brain areas. Unfortunately, three of five studies reported effects such as correlations of peak voxels signal and behavioural results [33,34,59], rendering these results non-independent and hard to interpret in terms of actual effects [58]. In another study [55], low relevance weighted prediction error in the salience task was associated with high activation in the hippocampus, which was deemed not in line with previous results [81], where large prediction errors are associated with hippocampal response. This unexpected result was interpreted as an effect of higher-order processes regulating memory encoding when the prediction error is low and memorizing is adaptive [55]. Winton–Brown and colleagues [27] found that reward-induced connectivity between ventral striatum/pallidum and midbrain was significantly greater in UHR than in HC. In addition, greater connectivity was correlated with unusual thought content in UHR. The authors interpreted these results as evidence of a disrupted circuitry, where enhanced hippocampal activity influences ventral striatum, pallidum, and midbrain functioning [82]. These findings tentatively suggest the engagement of disrupted ventral striatum and hippocampus processing in aberrant salience tasks.

The other major brain system is the salience network, primarily associated with higher cortical structures, insulae, and anterior cingulate cortex. The salience network is seen as a network whose function is to detect and “sort” salient stimuli [83] and to regulate and switch between other, task-oriented, and default mode networks [84]. It is implicated as the source of disrupted neural processing and symptoms of psychosis in the aberrant proximal salience concept [85]. Noteworthy, recent findings point to the role of aberrant metabolism of glutamate in regulation of the anterior cingulate functions [86] and dysregulation of the salience network in schizophrenia [74]. On the symptomatic level, disrupted salience network processing, along with its disrupted connectivity to task-positive and default mode networks, is associated with cognitive deficits, reality distortions [87], and hallucinations [88,89]. It is further hypothesized that the processing of the salience network is associated with aberrant attribution of salience and meaning to stimuli [88], which is a key process in aberrant salience cognitive bias. At the same time, it is associated with diminished capacity to differentiate between internal and external states [87,90], which is key in the source monitoring cognitive bias. 

In the current review, Walter and colleagues [36] reported an increased bold response in the left insula in the schizophrenia group with high positive symptoms, compared to the group with low positive symptoms. Additionally, response in the left insula correlated negatively with medication dose. Another study [56] found differences in bold response to adaptive salience processing in the anterior cingulate and right hemisphere insula cortices between FEP patients and healthy controls. Additionally, in FEP patients without medication, there was a negative correlation between right hemisphere insula activation during adaptive salience and hallucinations score. Reviewed studies showed differences in the regions of the salience network in individuals with psychosis on the static contrast estimates. However, Smieskova and colleagues [56] and others point out that future research of salience network in psychosis may benefit from adopting a connectivity-based approach, allowing to study more intricate functional disruptions in this network [87,88]. Current findings suggest the engagement of disrupted insula and anterior cingulate cortices processing in the aberrant salience tasks.

To sum up, the role of excessive release of dopamine in psychotic disorders is well documented [91,92], and associated with predictive processing account [23]. Notably, both these systems, the cortico-striatal-thalamic circuitry and the salience network, are at least to a degree modulated by dopaminergic [23,85,93] and glutamatergic [94] functioning. Reviewed studies [23,94] indicate that individuals with psychotic disorders and at risk of psychosis have altered salience processing, which is associated with disrupted functioning of the subcortical dopaminergic pathways and salience network, in comparison to healthy controls [23,72]. However, the heterogeneity of the studied populations and methods, methodological issues, like reporting correlations for peak voxels, and most notably variability and inconsistency in results, means that more research is needed to replicate these findings reliably.

### 4.3. Behavioral Results for Source Monitoring 

Studies considered in the current review show that the most common source monitoring paradigm is the external–internal paradigm, which was implemented in ten out of eleven studies. Only one study employed the external–external source monitoring paradigm [70]. In most of the studies, patients with schizophrenia spectrum disorder were compared to a non-clinical control group. Two papers [60,61] described a population of schizophrenia participants divided into groups with and without hallucinations. Despite the fact that most of the studies used an external–internal paradigm, the stimulus and task design usually differed. Nonetheless, a substantial part of the reviewed articles revealed significant differences between patients with SSD and non-clinical control groups in the source monitoring task performance. The obtained results are in line with previous meta-analyses [3,41]. 

Ten studies implemented the external–internal paradigm. Most of them reported significant differences in the overall source monitoring performance [60,61,62,66,67,68], where patients with schizophrenia made more misattributions than the control group. Significant effects for overall source monitoring task accuracy varied from medium (*d* = 0.66) [66] to large (*d* = 0.98) [63]. Thoresen et al. [64] showed that patients made more misattributions where they classified the imagined actions as viewed with a very large effect size (*d* = 1.30). Moreover, Vinogradov et al. [66] demonstrated that the schizophrenia group had a significantly lower hit rate for self-generated items than the healthy controls (*d* = 3.04). However, in this case, the effect size should be interpreted with caution due to the considerably small sample size. In the study by Kambeitz–Ilankovic et al. [69], there was a significant difference between FEP and HC group in the self-speech conditions while it was preceded by an invalid cue (picture of another person’s face) with a large effect size (*d* = 0.8). On the other hand, one study [70] that implemented the external–external source monitoring paradigm has not found significant differences in the task performance between groups.

To sum up, for the internal–external source monitoring paradigm, the results appear to be in the direction of previous meta-analyses [3,5,41], showing that patients with schizophrenia have a general tendency to commit more misattributions in source monitoring tasks. Contrarily, the same effect has not been observed in the study [70] that investigated the external–external discrimination paradigm. Therefore, in the current review, the results for this type of discrimination are inconclusive since the considered paradigm is substantially underrepresented. Thus, more research is needed to add new insight into the topic. Moreover, most of the above studies considered patients with schizophrenia and one with the first episode of psychosis. Thus, more research is needed on ultra-high risk of psychosis samples to exclude the potential impact of medication, impaired general cognitive functioning in schizophrenia patients, motivational aspects, and other factors that could potentially influence performance in all sorts of different behavioural tasks in patients with schizophrenia [95,96]. 

The current review provides additional evidence to already existing knowledge on source monitoring characteristics across psychotic patients. In future studies, direct comparison in source monitoring performance between schizophrenia patients and participants without full-blown psychotic symptoms is needed, as it would widen the knowledge on source monitoring deficits across the continuum for psychosis.

### 4.4. Neuroimaging Results for Source Monitoring

The reviewed studies used a variety of contrasts in the neuroimaging analyses. Some of the studies concentrated on the effects concerning the difference between reality monitoring and a control task [62], other on more specific effects associated with a type of memory or stimuli source [42,64], and there were also studies which searched for effects associated with misidentifying memory source [70]. This heterogeneity in sought out effects may translate into the variability of obtained brain activations. Nevertheless, we were able to identify some of the neural effects that were shared across different studies, like the medial prefrontal cortex (mPFC), hippocampus, middle and superior temporal gyri, and anterior cingulate cortex (ACC). 

The brain structure most often identified in the found studies was the medial prefrontal cortex [62,63,66,67,68]. In the majority of cases, this part of the prefrontal lobes was under investigation as a region of interest, as it was implicated in the studies of self-referential and memory processing [65,97] and reality monitoring in healthy subjects (for review on reality monitoring see: [98]). Most of the studies point to diminished activity in the mPFC in patients during contrasts associated with source monitoring, except a study by Kumari and colleagues [63], where patients with worse results in the verbal monitoring task had higher activations in mPFC than patients with better results. Wang and colleagues [68] used mPFC as a seed in a connectivity analysis. They found a stronger relationship between the medial frontal cortex and superior temporal gyrus in individuals with schizophrenia during self-monitoring condition relative to controls. The authors interpreted this result as a brain circuitry abnormality that contributes to the self–other confusion, which later manifests as positive symptoms. Taken together, findings from these studies suggest that diminished activity of the mPFC is associated with the disrupted source monitoring, facilitating confusion in attributing the source of perceptions or memories from the self to others and contributing to the “externalization bias” [10]. 

The current review also implicated the hippocampus in the source monitoring deficits. The study by Thorensen and colleagues [64] found diminished activity in this region during source monitoring of imagined stimuli. Additionally, diminished hippocampal activity was associated with higher levels of delusions. Kumari and colleagues [63] found greater response in the hippocampus during source monitoring of self-generated stimuli in the patient’s group with better results in this task. Medial parts of the temporal lobes are associated with memory encoding and retrieval [99,100], which is a crucial aspect of the source monitoring processes. For an extensive review of the role of the hippocampus in the source monitoring, see [19]. One of the recent findings points out that mPFC modulates the hippocampus’s functioning, minimising interference that would otherwise impede discrimination between similar memory representations [101]. This could suggest an impaired top–down circuitry in schizophrenia, where disruption of mPFC affects the functioning of the hippocampus during memory tasks.

Several of the reviewed studies also found effects in the superior and middle temporal gyri. This was expected in the context of the majority of the found studies using speech-based stimuli. Effects of source monitoring were primarily associated with diminished response in these parts of the temporal cortex [42,61,63,69]. Also, Wang and colleagues [68] and Mechelli and colleagues [61] found a disrupted connectivity pattern between the superior temporal gyrus and the mPFC and ACC, respectively. These temporal regions are associated with sensory processing and cross-modal integration of stimuli, but also self-referential processing and discrimination of source of speech [42,61,63,69]. A similar function of reality monitoring is associated with the paracingulate sulcus. This tertiary structure is morphologically diverse in the human population and located between the mPFC and the ACC [102,103]. In the current review, apart from studies concentrating on the mPFC, three studies found significant effects in the proximal vicinity of the paracingulate gyrus, namely in the anterior cingulate cortex [60,61,65]. However, these results are heterogeneous, showing diminished ACC activity in the patients in one study [60] and increased activity in another [65] during the processing of self-generated stimuli. Additionally, Mechelli and colleagues reported diminished connectivity between the ACC and the superior temporal gyrus in hallucinating patients. Nonetheless, these results suggest that structures placed in the broad anterior cingulum area of the brain are functionally relevant to the source monitoring bias [98].

Noteworthy, several studies reported effects localized in the basal ganglia, i.e., the striatum and the putamen [63,65,67]. These structures were not extensively discussed in the literature on source monitoring. Subramaniam and colleagues [67] noted that perhaps in patients with schizophrenia, there is a disruption in a dopaminergic frontostriatal pathway associated with motivational and goal-directed behaviour. It may be suggested that functional abnormalities of the basal ganglia, especially striatum, is a reflection of disrupted predictive processing during the source monitoring. This would coincide with aberrant salience account, where striatum is modulated by the excessive hippocampus activity, but further studies are warranted to elaborate on the role of basal ganglia in disruptions of source monitoring processes.

To sum up, schizophrenia patients show disruptions in several brain regions during source monitoring tasks. The obtained results are in line with proposed neural correlates of reality monitoring in the context of the mPFC, medial temporal lobe, and paracingulate gyrus and the ACC [98]. Also, significant between-group differences during source monitoring were found for middle and superior temporal gyri. Notably, these effects are also consistent with neural correlates of hallucinations in schizophrenia, underlying the role of disrupted source monitoring in positive symptoms of psychosis [104].

### 4.5. Overlaps in Neuroimaging Effects

The review of the found studies pointed to two key brain regions implicated in both aberrant salience and source monitoring biases. The first region is the hippocampus, which is associated with memory processing [105] but also the regulation of the dopamine functioning in the mesolimbic pathways [82,106]. The other one was the anterior cingulate cortex, which is a key structure in the brain salience network, responsible for regulating internal states and self-referential processing [73,107]. Figure 2 demonstrates the most common functional differences between patients and controls in aberrant salience and source monitoring biases. Additionally, it marks overlaps and specific effects for functional correlates of aberrant salience and source monitoring.

Evidence points to the hippocampus as one of the key structures responsible for regulating dopamine functioning [108,109,110]. Metabolism of this neurotransmitter is disrupted in schizophrenia [92,111] and states of clinical risk of psychosis [23,112] and is associated with positive symptoms of psychosis [82]. In addition, studies are pointing to the structural and functional disruption in the hippocampus in schizophrenia and the risk of psychosis [102,105,110,113]. States of psychosis risk and transition to psychosis are associated with structural and functional abnormalities in the medial temporal cortex, i.e., the hippocampus [114,115,116]. Additionally, schizophrenia is associated with abnormality in the hippocampal function [113,117]. A study on schizophrenia patients’ brain response to the dopamine-antagonist medication showed that it affected hippocampus and ventral striatum in the early stages of treatment and that normalization of hippocampal function was associated with a better response to treatment [118]. On the other hand, there are accounts linking dopaminergic dysfunction in the hippocampus with disrupted metabolism of glutamate [94,119]. Lodge and Grace [120] described the hippocampus as a focal point of influence of various risk factors, like substance abuse, genetic risk, or psychological stress, contributing to the development of psychotic symptoms.

Studies in the current review are in line with these results, showing disruption of hippocampus functioning in patients with schizophrenia and risk of psychosis when exhibiting aberrant salience and disrupted source monitoring. However, in both these cognitive biases, the hippocampus is seen as a part of two partially distinct circuitries associated with different cognitive processes. In aberrant salience, it is seen as a regulator of dopaminergic neurons in the ventral striatum and the midbrain [27,33,34,82]. In source monitoring, it is seen as a part of circuitry engaged in recollection and source identification of memories, where prefrontal regions, associated with self-referential processing and cognitive control, have the regulating role [19,64,98]. There is also light to be shed on the relationship between the hippocampus and the basal ganglia during source monitoring. To sum up, despite its diverse functions, the hippocampus seems to be a key brain structure in aberrant salience and disrupted source monitoring in individuals with schizophrenia and at risk of psychosis.

Another structure associated with both cognitive biases was the anterior cingulate cortex. Studies show that disruption of function and structure of the anterior cingulate cortex is associated with psychosis [121,122,123,124]. The high risk of psychosis and transmission to psychosis is associated with reduced grey matter in the anterior cingulate cortex [99]. A study on patients’ response to dopamine-antagonist treatment showed an effect in the anterior cingulate and medial prefrontal cortex [100]. Recent findings point to the association between disrupted glutamate metabolism and salience network functioning [74]. Additionally, it should be noted that differences in functional outcomes in the ACC may be originating in nonspecific grey matter loss [125] or disrupted neurovascular coupling in this region [126]. 

The current review is in line with studies linking ACC functioning with psychosis. We demonstrated that both aberrant salience and source monitoring are linked to abnormal functioning of this brain structure. However, in aberrant salience studies, it is interpreted in the context of disrupted processing of the brain salience network [55,85]. In this approach, the salience network is responsible for regulating cognitive processing and reallocation of attention, switching between task-positive and task-negative brain networks. In the source monitoring studies, observed disruptions in the ACC functioning are identified as associated with the dysfunction of the paracingulate sulcus, primarily as it seems due to the anatomical proximity and relationship [102]. This argument appears to be pervasive due to the association of paracingulate sulcus, reality monitoring [98,103], and hallucinations [127,128].

To sum up, both brain regions identified in this review as associated with both aberrant salience and disrupted source monitoring are discussed in their respective bodies of literature in the context of various neural mechanisms, corresponding to their cognitive and phenomenological mechanisms and character. This apparent dissociation calls for an attempt in bridging these two cognitive biases into an account that will encompass their characteristics and enable future studies on their possible joint functional neural mechanisms.

### 4.6. Integration of Results on Aberrant Salience and Source Monitoring

A number of studies and reviews have discussed the possible integration of aberrant salience and source monitoring [22,43,46,129]. However, our study is the first to systematically review neural correlates of aberrant salience and source monitoring in functional MRI studies. Our findings seem to be in line with these previous accounts. Observed functional differences between individuals with schizophrenia or the risk of psychosis during source monitoring and aberrant salience may indicate a disturbance in hierarchical predictive processing and disturbances in minimal self [43,46].

Observed possible overlaps between functional neural correlates of these cognitive biases suggest that they may function in a feedback loop. In such an account, aberrant salience and disrupted source monitoring lead to an experience of externalization and emergent experience of aberrant attributional processes of cognitive functions and one’s own experiences as originating externally. Aberrant salience is associated with strong priors or underweighted prediction errors [23]. They lead to erroneous prediction updates, which in turn may lead, on the next hierarchical level of predictive processing, to disrupted priors in processes of source monitoring and attribution [43]. On a phenomenological level, aberrant salience translates to a sense of heightened yet erratic motivational and attentional functioning. A state of perceptual and self-referential catch-as-catch-can and disrupted attentional control [47]. As Mitchell and Johnson [19] noted in their account of cognitive mechanisms of source monitoring, memory is selective and contextual, influenced by meta-memory, knowledge, schemas, biases, and temporarily elicited states. In a state of aberrant salience, inferences about the origin of perceptual and cognitive phenomena are burdened simultaneously with erroneous predictions and with disrupted self-referential processing and stimuli discrimination, leading to a disturbed sense of ownership of cognitive processes and agency. In this approach state of aberrant salience would be deemed a context-modifier for more meta-level processes of source monitoring. Nevertheless, their interaction would be necessary to produce an external attribution of internal phenomena, like sensory experiences or thoughts. This attribution, resulting in a qualitative change of how reality is perceived, could lead in turn to positive symptoms of psychosis [130,131].

However, as we assume a hierarchical type precedence of aberrant salience over source monitoring deficits [43], we cannot certainly presuppose temporal precedence of these two cognitive biases. Indeed, there are studies indicating the stronger predictive potential of source monitoring deficits in early psychosis’ self-disturbances than that of aberrant salience [46,132]. Our assumption is supported by neural correlates of researched cognitive biases, where source monitoring is associated more with higher-order cortical structures. Nonetheless, an account where a state of aberrant salience occurs earlier and contributes to the source monitoring disruptions warrants further investigation, especially with the use of neuroimaging studies.

## 5. Conclusions

To sum up, we conducted a systematic review of fMRI studies on source monitoring and aberrant salience. There are several areas of concern or questions that could be addressed by future research. The first one is the reliability of available fMRI studies of aberrant salience, where the majority of expected effects were non-significant. A systematic review of behavioural tasks of aberrant salience is warranted, and future neurocognitive studies of this cognitive bias should employ larger sample sizes. Additionally, as most of the studies on aberrant salience concentrate on dopamine signalling, future studies could benefit from integrating this perspective with accounts on the role of glutamate in salience processing. Another question is about neural correlates of source monitoring in participants with high clinical risk of psychosis. To our knowledge, there are no fMRI studies that would elucidate neural correlates of disrupted source monitoring in such a sample. 

Furthermore, we found that several studies of source monitoring found effects associated with basal ganglia, yet the role of this structure was not extensively discussed in the previous literature. Future accounts of source monitoring could explore how disrupted motivational and salience processing adds to disrupted source monitoring. This could also be an incentive to explore the role of predictive processing in the source monitoring accounts. Additionally, we found overlap in the functional correlates of aberrant salience and source monitoring, namely the anterior cingulate cortex. However, effects found in this structure in the source monitoring accounts are associated with morphological diversity of the paracingulate sulcus, which is responsible for self-monitoring functions. Additionally, the ACC region is associated with other types of physiological abnormalities in psychosis. Future research could shed additional light on how specific structures in the anterior cingulate area contribute to aberrant salience and source monitoring.

Finally, we reviewed and discussed neural correlates of aberrant salience and source monitoring in patients with schizophrenia or at clinical risk of psychosis. We point to several regions whose functioning is associated with aberrant salience—ventral striatum and the insula cortex—and with source monitoring—the medial prefrontal cortex and superior and middle temporal gyri. There was a possible functional neural overlap in the anterior cingulate cortex and paracingulate sulcus and hippocampus. Our review indicates that aberrant salience and source monitoring may share neural mechanisms, suggesting their joint role in producing disrupted external attributions of perceptual and cognitive experiences, thus elucidating their role in positive symptoms of psychosis. Future studies, both behavioural and fMRI, may benefit from investigating source monitoring and aberrant salience simultaneously. This may provide empirical verification of similarities and functional overlaps of these two important cognitive biases that contribute to psychosis.

## Figures and Tables

**Figure 1 jcm-10-04126-f001:**
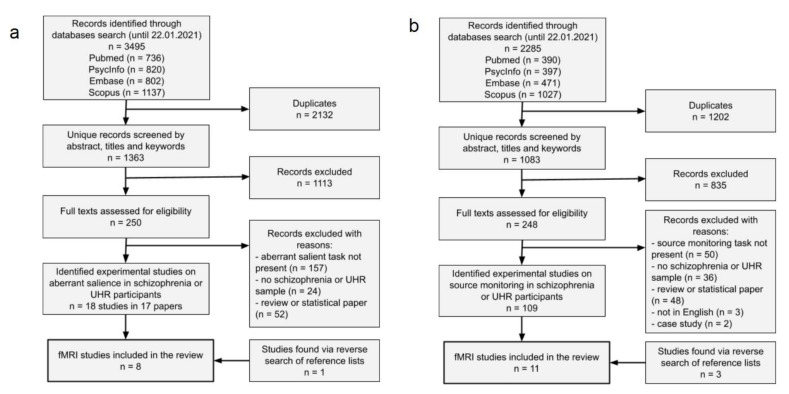
Flow diagram of literature search and review for aberrant salience (**a**) and source monitoring (**b**) studies.

**Figure 2 jcm-10-04126-f002:**
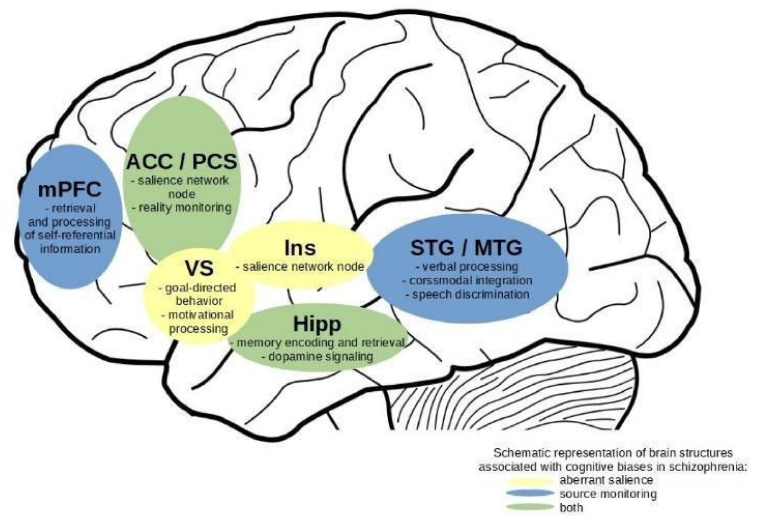
Schematic representation of brain regions with effects detected between schizophrenia patients and non-clinical controls in fMRI studies of aberrant salience and source monitoring. For display purposes, effects are presented only on the left hemisphere of the brain. mPFC—medial prefrontal cortex, ACC/PCS—anterior cingulate cortex and paracingulate sulcus, vs.—central striatum, Ins—insula, Hipp—hippocampus, STG/MTG—superior and middle temporal gyri.

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
