# Peer review of "Neural Correlates of Aberrant Salience and Source Monitoring in Schizophrenia and At-Risk Mental States—A Systematic Review of fMRI Studies"

_jcm, 2021, doi:10.3390/jcm10184126_

Round 1
Reviewer 1 Report
I enjoyed reading this work. I recommend publishing this pending includion of missing literature as well as rewriting the emphasis given to certain statements, where evidence does not hold up.
The major challenge is the overinterpretation of literature (often the authors say neuronal overlap, etc., when there is no microstructural or cellular evidence; this needs to be toned down)
In some places the authors are overstating the inferences that could be drawn from their reading of the literature: e.g. "Evidence points to the hippocampus as the structure responsible for regulating dopamine functioning" - single brain regions are not responsible for a neurotransmitter in this way. In fact hippocampus has more established role in pattern separation - see some recent relevant findings Hippocampus 2020 Oct;30(10):1058-1072.
The authors are equating Salience Network function to dopamine excess. I don't think this has been conclusively demonstrated. In fact, more evidence exists for Glutamate - Salience Network link: see Biological Psychiatry 2020 Aug 1;88(3):273-281 and Antioxidants 2021 Jan 8;10(1):75.
How are symptoms linked to the discussed deficits? There is a lot on brain structure and symptoms of the networks discussed here:
I am surprised by exclusion of thalamus in its role of salience and source detection (attentional) functions: in terms of relevance to schizophrenia see Am Acad Child Adolesc Psychiatry. 2021 Apr;60(4):479-489.
Cingulate system may have 2 different physiological defects , as discussed here: this requires further discussion when interpreting fMRI studies. Front Psychiatry. 2020 Aug 5;11:754. Also see the role of ACC in defects unrelated to source monitoring/salience detection e.g. Journal of Psychiatry & Neuroscience, 2021 Apr 27;46(3):E337-E346.
Author Response
Rev 1.:
I enjoyed reading this work. I recommend publishing this pending includion of missing literature as well as rewriting the emphasis given to certain statements, where evidence does not hold up.
Thank you for the positive feedback and the opportunity to address your remarks!
The major challenge is the overinterpretation of literature (often the authors say neuronal overlap, etc., when there is no microstructural or cellular evidence; this needs to be toned down)
Thank you for this remark. We specified wherever possible that we meant functional neural overlaps, understood as activation of synonymous brain regions. We also added more modal adjectives and adverbs in the appropriate parts of the discussion section to emphasize the hypothetical character of some of our conclusions.
In some places the authors are overstating the inferences that could be drawn from their reading of the literature: e.g. "Evidence points to the hippocampus as the structure responsible for regulating dopamine functioning" - single brain regions are not responsible for a neurotransmitter in this way. In fact hippocampus has more established role in pattern separation - see some recent relevant findings Hippocampus 2020 Oct;30(10):1058-1072.
We corrected the sentence with this undue shorthand. Additionally, we scanned the manuscript in search of similar overstatements. Thank you for noticing!
The authors are equating Salience Network function to dopamine excess. I don't think this has been conclusively demonstrated. In fact, more evidence exists for Glutamate - Salience Network link: see Biological Psychiatry 2020 Aug 1;88(3):273-281 and Antioxidants 2021 Jan 8;10(1):75.
The Reviewer probably refers here to the first paragraph of the discussion of fMRI findings in aberrant salience. We reworked this sentence to be less ambiguous, as we wanted to point out some anatomical overlap between subcortical dopaminergic pathways and the Salience Network. The aim of taking note of this overlap was to bring readers’ awareness to how activations in these structures may be interpreted and how it was done in our paper.
On the other hand, the account of glutamate regulating the Salience Network seems very interesting and timely. We added to the discussion some passages related to results brought up by the Reviewer. Thank you for directing our attention to this area of research.
How are symptoms linked to the discussed deficits? There is a lot on brain structure and symptoms of the networks discussed here:
Thank you for this question. However, we are not sure if there isn’t a piece of text missing? But to answer a general comment on the link between neuroimaging effects and symptoms, we feel that the discussion section suitably addresses this issue, with references to previous studies showing associations between neuroimaging, biases and symptoms. Our central premise is that interaction between aberrant salience and source monitoring deficits could be crucial for disrupted processing of self, the process of deeming internal experiences as originating externally and positive symptoms of psychosis.
I am surprised by exclusion of thalamus in its role of salience and source detection (attentional) functions: in terms of relevance to schizophrenia see Am Acad Child Adolesc Psychiatry. 2021 Apr;60(4):479-489.
Only one study found differences in thalamic activity between clinical and control groups in the source monitoring paradigms in the current review. As for aberrant salience, e.g. Schmidt et al., 2016 or Smieskova et al., 2015 reported main effects associated with adaptive salience processing in the thalamus, but no effects of interest (group differences) were found. So perhaps we could say it is more of a lack of systematic findings rather than exclusion. As we acknowledge the role of the thalamus in various types of cognitive processing, we feel that the review in its current form is sizable, and we decided not to pursue this topic.
Cingulate system may have 2 different physiological defects , as discussed here: this requires further discussion when interpreting fMRI studies. Front Psychiatry. 2020 Aug 5;11:754. Also see the role of ACC in defects unrelated to source monitoring/salience detection e.g. Journal of Psychiatry & Neuroscience, 2021 Apr 27;46(3):E337-E346.
Thank you for directing us to this novel research. We modified the discussion section to accommodate these findings.
Reviewer 2 Report
The present work systematically reviews aberrant salience and source monitoring fMRI studies, allowing to shed new light on the complex relation between these two constructs. The findings and the conclusion that are drawn are important, so that I believe this could be a very interesting work.
Some major and minor points could be addressed to better define the message the Authors want to deliver and strengthen the structure of the paper:
1 Line 56: On a perceptual level, aberrant salience is associated with strong prior predictions or underweighted prediction error, which leads to excessive reliance on priors and undervaluation of sensory input [23].
This is an interesting and complex point. Many works reported weak priors and these contrasting findings are still hard to interpret (https://www.sciencedirect.com/science/article/pii/S0006322318315324). If the Authors use predictive coding to explain aberrant salience, they should better integrate the cues offered by this theory.
2 METHODS: The review seems well-conducted. However, I would check up at least the PRISMA guidelines https://www.bmj.com/content/372/bmj.n71 and report a flowchart.
3 I would have expected other measures to be included as correlates of Aberrant Salience, such as oddball tasks (https://pubmed.ncbi.nlm.nih.gov/32710172/ https://pubmed.ncbi.nlm.nih.gov/25037525/). Could you please explain why these studies were not included?
4 RESULTS: this review accurately reports results. However, the reader can be puzzled by the amount of information in both the text and the tables. I believe that a cleaner schematization of the findings and being more synthetic would greatly improve the readability of the present work. Parallelly, information can be added to the figure (for instance, about the number of studies finding or not between-groups differences in each area). Another aspect that could be reported more clearly is how fMRI measures are related to behavioral findings in HC vs CHR vs SSD in the various studies.
MINOR COMMENTS
1 Line 61: However, this model is still developed, with important questions yet to be answered. Maybe the Authors wanted to word this sentence as “is still in need to be developed” or similar.
2 Line 175: Four à Fourth
3 Line 260: the meaning of Adaptive Salience should be introduced.
4 DISCUSSION: the Authors report that i) few significances for Aberrant Salience are documented. ii) most of the structures they identify are related to Salience Network and Default Mode Network. iii) basically, all the reported structures pertain to the same anatomical dopaminergic system.
The bulk of evidence questions to what extent aberrant salience and source monitoring are separate domains. The Authors distinguished between different phases of stimuli processing, and it would be interesting to deepen this aspect including more findings.
Our group recently proposed parallel perspectives concerning these aspects (https://pubmed.ncbi.nlm.nih.gov/32208349/), there is no need to include the work in the discussion but I share the link as the topic is significantly related to the Authors’ review.
Author Response
Rev 2.
Comments and Suggestions for Authors
The present work systematically reviews aberrant salience and source monitoring fMRI studies, allowing to shed new light on the complex relation between these two constructs. The findings and the conclusion that are drawn are important, so that I believe this could be a very interesting work.
Thank you for the positive feedback and the opportunity to address your remarks!
Some major and minor points could be addressed to better define the message the Authors want to deliver and strengthen the structure of the paper:
1 Line 56: On a perceptual level, aberrant salience is associated with strong prior predictions or underweighted prediction error, which leads to excessive reliance on priors and undervaluation of sensory input [23].
This is an interesting and complex point. Many works reported weak priors and these contrasting findings are still hard to interpret (https://www.sciencedirect.com/science/article/pii/S0006322318315324). If the Authors use predictive coding to explain aberrant salience, they should better integrate the cues offered by this theory.
We added a suitable passage in the introduction to emphasize the ongoing debate on the predictive coding account in psychosis. However, we think that the same group of authors we refer to are weighting(nomen omen) in their more recent works towards the greater role of strong priors in the formation of positive symptoms, especially when integrated with the aberrant salience account (especially https://www.sciencedirect.com/science/article/pii/S0006322320313834, but also https://www.sciencedirect.com/science/article/pii/S1364661318302821). Thank you for pointing out the complexity of the issue. Nevertheless, it would be probably out of the scope of our review on cognitive biases to discuss conflicting evidence on predictive coding and even attempt at resolving this discussion.
2 METHODS: The review seems well-conducted. However, I would check up at least the PRISMA guidelines https://www.bmj.com/content/372/bmj.n71 and report a flowchart.
We provided flowcharts for studies selection in the methods section. As our review was conducted in accordance with the PRISMA guidelines we also added this information in the methods section. We also added assessment of studies quality to the supplementary material.
3 I would have expected other measures to be included as correlates of Aberrant Salience, such as oddball tasks (https://pubmed.ncbi.nlm.nih.gov/32710172/ https://pubmed.ncbi.nlm.nih.gov/25037525/). Could you please explain why these studies were not included?
Thank you for this question! We debated it in our team when preparing the first version of the manuscript. We aimed at reviewing studies employing experimental paradigms which tap directly into the construct of aberrant salience, i.e. attribution of meaning which is not valid or justified in a given context (e.g. reacting to irrelevant cues as they were associated with reward). Because of that, we chose to concentrate on paradigms that manipulate or blind relevance. Reactions to oddballs in oddball tasks, in our opinion, could be more conceptualized as a correlate of distraction inhibition or sustained attention. These processes are key for understanding saliency processing but are not a per se operationalization of aberrant salience.
4 RESULTS: this review accurately reports results. However, the reader can be puzzled by the amount of information in both the text and the tables. I believe that a cleaner schematization of the findings and being more synthetic would greatly improve the readability of the present work. Parallelly, information can be added to the figure (for instance, about the number of studies finding or not between-groups differences in each area). Another aspect that could be reported more clearly is how fMRI measures are related to behavioral findings in HC vs CHR vs SSD in the various studies.
Thank you for having a care for the readability of our manuscript! We slightly edited the tables to be more easily readable and comprehensible. However, as for the more synthetic presentation of the results of our review, we deliberately aimed at reviewing and presenting gathered data in their fullness. We put effort into systematically describing the findings, creating a narrative and now hope that the text will draw readers' attention. However, we could rework it to fit into a more simplified way of data presentation if the Reviewer finds the present version of the manuscript still potentially puzzling for the readers.
MINOR COMMENTS
1 Line 61: However, this model is still developed, with important questions yet to be answered. Maybe the Authors wanted to word this sentence as “is still in need to be developed” or similar.
Thank you for pointing that out. The verb here should be in the continuous form, so we changed part of this sentence to “still being developed”.
2 Line 175: Four à Fourth
This was corrected.
3 Line 260: the meaning of Adaptive Salience should be introduced.
We provided a definition of adaptive salience at the beginning of his paragraph.
4 DISCUSSION: the Authors report that i) few significances for Aberrant Salience are documented. ii) most of the structures they identify are related to Salience Network and Default Mode Network. iii) basically, all the reported structures pertain to the same anatomical dopaminergic system.
The bulk of evidence questions to what extent aberrant salience and source monitoring are separate domains. The Authors distinguished between different phases of stimuli processing, and it would be interesting to deepen this aspect including more findings.
Thank you for deeming this part of our discussion interesting! Firstly, it should be noticed that according to Reviewer’s 1. remarks, we toned down the simplifications and overstatements regarding the dopaminergic system. Secondly, unfortunately, to our knowledge, the temporal relation of aberrant salience and disrupted source monitoring in perceptual processing of individuals with or at risk of psychosis is not yet well researched. We referred in the discussion to the recent work of prof. Barnaby Nelson and other researchers dealing with the concept of minimal self-disturbance. Their initial, cross-sectional results suggest that our conceptualization may be premature. Disrupted source monitoring may be a more core (and thus, at least hypothetically by this point, temporally preceding) aspect of the minimal self-disturbance in psychosis than aberrant salience. Because of that, we feel that deepening our account could be too speculative.
Our group recently proposed parallel perspectives concerning these aspects (https://pubmed.ncbi.nlm.nih.gov/32208349/), there is no need to include the work in the discussion but I share the link as the topic is significantly related to the Authors’ review
Thank you for sharing your work on the topic! We will certainly delve into it!
Round 2
Reviewer 1 Report
none